# Synthesis, Thermal Properties and Curing Kinetics of Hyperbranched BPA/PEG Epoxy Resin

**DOI:** 10.3390/polym11101545

**Published:** 2019-09-23

**Authors:** Tossapol Boonlert-uthai, Chavakorn Samthong, Anongnat Somwangthanaroj

**Affiliations:** Department of Chemical Engineering, Faculty of Engineering, Chulalongkorn University, Bangkok 10330, Thailand; tossapol.bu@gmail.com (T.B.-u.); tao_hp_@hotmail.com (C.S.)

**Keywords:** hyperbranched epoxy, curing kinetics, polyethylene glycol, thermal properties

## Abstract

The hyperbranched epoxy resins (HBE) composed of bisphenol A (BPA) and polyethylene glycol (PEG) as reactants and pentaerythritol as branching point were successfully synthesized via A_2_ + B_4_ polycondensation reaction at various BPA/PEG ratios. The ^13^C NMR spectra revealed that the synthesized HBE mainly had a dendritic structure as confirmed by the high degree of branching (DB). The addition of PEG in the resin enhanced degree of branching (DB) (from 0.82 to 0.90), epoxy equivalent weight (EEW) (from 697 g eq^−1^ to 468 g eq^−1^) as well as curing reaction. Adding 5–10 wt.% PEG in the resin decreased the onset and peak curing temperatures and glass transition temperature; however, adding 15 wt.% PEG in the resin have increased these thermal properties due to the lowest EEW. The curing kinetics were evaluated by fitting the experimental data of the curing behavior of all resins with the Šesták–Berggren equation. The activation energy increased with the increase of PEG in the resins due to HBE’s steric hindrance, whereas the activation energy of HBE15P decreased due to a large amount of equivalent active epoxy group per mass sample. The curing behavior and thermal properties of obtained hyperbranched BPA/PEG epoxy resin would be suitable for using in electronics application.

## 1. Introduction

Epoxy thermoset has become the most recognized adhesive and is widely used in demanding industries such as aerospace, automotive, building and construction, and electrical and electronics industries. For the electrical and electronics industries, especially hard disk drive (HDD) production, thermoset epoxy has been used to adhere the important parts, such as head gimbal assembly (HGA) used for reading and writing the digital data on a disk in HDD. The physical, thermal, mechanical, thermomechanical, and rheological properties of the thermoset epoxy used in HDD should be investigated in order to match the HGA process in HDD production. Generally, the commercial epoxy adhesive is based on diglycidyl ether of bisphenol A (DGEBA) because of high thermal and mechanical properties, good weather and chemical resistances, low shrinkage, and high adhesion strength [1,2,3]. However, the unmodified epoxy has some disadvantageous properties, i.e., inherent brittleness and low toughness, limiting its utilization for the advanced applications which require high mechanical performance. Thus, the modification of epoxy by an incorporation of toughening agent and plasticizer was investigated and the natures of these fillers are of great importance affecting the final properties of the cured epoxy-based adhesive products.

In the present, hyperbranched polymers are novel three-dimensional macromolecules and are produced in a one-step procedure by multiplicative growth from a multi-functional core to form the repeated branching units via polycondensation of AB_x_ monomers [4,5,6,7,8,9]. If x ≥ 2 and functional group of A molecule reacts only with functional group of B molecule, the production of highly branched polymers is ensured. Hyperbranched epoxy resins are easy to synthesize, have low viscosity, high solubility and large number of end functional groups; therefore, they are widely produced and developed for industrial scale productions and applications, such as oil field chemical, additive and resin for waterborne applications, dispersion agent, rheology modifier, and crosslinker for elastomer [9]. The most important feature of hyperbranched polymer is their degree of branching (DB) and branching factor classified into dendritic (D), linear (L), and terminal (T) units in the macromolecular structure [10].

Furthermore, glass transition temperature (T_g_) is the most important thermal property for a dendritic polymer. There is a relationship between DB and T_g_ [11,12,13,14]. It can be found that T_g_ gradually decreased with increasing DB. This can be explained by the fact that a greater degree of branching means more junction points and terminal units which lead to large numbers of free volume between branching chains, resulting in high flexibility [11,12,13,14,15,16]. Recently, there was a study to control epoxy equivalent weight (EEW) and performance of hyperbranched epoxy resins. It was found that T_g_ and mechanical properties of the cured hyperbranched epoxy/DGEBA composites were tuned by the EEW of hyperbranched epoxy resin and these properties were firstly increased and then decreased [17]. Moreover, it was found that the hyperbranched resin can act as toughener to enhance the mechanical property of the thermoset [18,19,20,21,22].

De and Karak [4,5] synthesized the hyperbranched epoxy resins by A_2_ + B_3_ and A_2_ + B_4_ polycondensation reactions between triethanol amine and in situ prepared diglycidyl ether of bisphenol A (DGEBA), and between pentaerythritol and in situ prepared DGEBA, respectively. It was found that the hyperbranched epoxy resin which was synthesized via one-step polycondensation and the aliphatic–aromatic moiety in the hyperbranched structure offer a high-performance toughened thermoset. Moreover, both reaction time and amount of B_4_ moiety affected DB, EEW, as well as thermal and mechanical properties. These properties were firstly enhanced and then reduced with increase in reaction time and the amount of B_4_ moiety. The properties of A_2_B_4_ hyperbranched epoxy thermoset were better than the properties of A_2_B_3_ hyperbranched epoxy thermoset, especially lower curing time and mechanical properties, such as toughness, elongation at break and adhesion strength.

There are several studies examining the addition of polyethylene glycol in the epoxy resins to improve low impact resistance of DGEBA by decreasing T_g_ of the thermoset [23,24,25,26]. However, in epoxy blended with low epoxy content, crystallization can occur when PEG content increases and it can decrease and hinder the cure reaction. In addition, if PEG is excessively added in the system, melting occurs, which reduces the thermal stability of the thermoset [26,27]. The physical properties of the cured epoxy depend on the structure of crosslinking network, curing temperature and curing time [28,29,30,31]. The relationship between network formation and final properties of the epoxy network is important in order to achieve the desired high-performance thermoset. The curing kinetics of the epoxy adhesive has been studied and analyzed via different techniques such as differential scanning calorimetry (DSC), dielectric relaxation spectroscopy, and gel permeation chromatography (GPC) [32].

This research aimed to obtain the epoxy adhesive whose properties are suitable for electronics application. The hyperbranched epoxy resins (HBE) were synthesized through A_2_ + B_4_ polycondensation reaction varying the ratios of aliphatic PEG and aromatic-containing bisphenol A (BPA) in A_2_ part. The chemical structures and degree of branching of the synthesized HBE resins were confirmed by FTIR, ^1^H and ^13^C NMR, and GPC. The curing behavior of HBE having different PEG fractions in their structures was also investigated by DSC. The curing kinetics was evaluated by fitting the experimental data with the auto-catalyzed reaction model.

## 2. Materials and Methods

### 2.1. Materials

Bisphenol A (BPA) used as A_2_ monomer for preparing the in-situ generated DGEBA monomer was purchased from Tokyo Chemical Industry Co., Ltd. (Tokyo, Japan) and was purified by recrystallization from toluene before using. Epichlorohydrin (ECH) as epoxidation reagent was obtained from Tokyo Chemical Industry Co., Ltd. Pentaerythritol (PE) used as B_4_ branch generating unit for reacting with the in-situ generated DGEBA was purchased from Tokyo Chemical Industry Co., Ltd., Japan and was purified by recrystallization from ethanol prior to use. Polyethylene glycol (PEG400, Mw = 400 g/mol) as A_2_ monomer was purchased from Sigma-Aldrich, St. Louis, MI, USA. Sodium hydroxide (NaOH) as base catalyst and sodium chloride (NaCl) were obtained from Ajax Finechem, Australia. Hydrobromic acid (HBr), acetic acid, potassium hydrogen phthalate, methyl violet, and chlorobenzene were used to determine epoxy equivalent weight (EEW) of the hyperbranched epoxy resins and were purchased from Tokyo Chemical Industry Co., Ltd., Japan. Diethylenetriamine (DETA) as amine curing agent was brought from Tokyo Chemical Industry Co., Ltd., Japan. Solvents were analytical grades and used as received.

### 2.2. Synthesis of HBE Resins

The hyperbranched epoxy resins were synthesized via the A_2_ + B_4_ polycondensation reaction [5] using BPA and PEG400 as A_2_ monomers and PE as B_4_ monomer (10 wt.% of total A_2_ monomer content). Mass ratios of BPA and PEG400 were varied at 100:0, 95:5, 90:10 and 85:15. The molar ratio of A_2_ monomer to ECH was fixed at 1:2. Five grams of BPA, 0.50 g of PE and 10.82 g of ECH were stirred using a magnetic bar in a two-necked round bottom-flask equipped with condenser and dropping funnel. When the mixture temperature was reached from room temperature to 60 °C, 5N aqueous NaOH solution (1.852 g equivalent to the hydroxyl groups) was added into the mixture drop wisely through the dropping funnel until the mixture temperature reached 110 °C (addition time was about 30 min). The reaction temperature was kept at 110 °C for 4 h. Afterwards, the mixture was immediately quenched in an ice bath to terminate the reaction and allowed to settle in a separation funnel. The organic layer was separated from the aqueous layer and purified by shaking with 15 wt.% NaCl solution followed by distilled water until its pH was 8–9. Eventually, the organic solution was dried under vacuum at 70 °C until the weight of the dried sample was constant. The synthesized resins were the viscous transparent liquids. The formulations for the synthesis of hyperbranched epoxy resins are tabulated in Table 1.

### 2.3. Preparation of Cured HBE

The synthesized HBE resins were homogenously mixed with DETA by mechanically stirring at room temperature for 10 min. The ratio of HBE to DETA was 1:1 molar ratio of active functional groups. The weight of amine curing agent of each system could be calculated by Equation (1). Epoxy Wt is weight of epoxy and phr amine is evaluated by Equation (2). Moreover, epoxy equivalent weight (EEW) of the mixture and NH-group equivalent can be calculated by Equations (3) and (4), respectively. Table 1 shows the composition of different formulations with respect to epoxy equivalent weight of resin.

(1)$$\mathrm{curing}\;\mathrm{agent}\;\mathrm{Wt}=\frac{\mathrm{epoxy}\;\mathrm{Wt}\times \mathrm{phr}\;\mathrm{amine}}{100}$$

(2)$$\mathrm{phr}\;\mathrm{amine}=\frac{\mathrm{NH}\;\mathrm{equivalent}}{\mathrm{EEW}}\times 100$$

(3)$$\mathrm{EEW}\;\mathrm{of}\;\mathrm{mixture}=\frac{\mathrm{Total}\;\mathrm{Wt}}{\frac{{\mathrm{Wt}}_{a}}{{\mathrm{EEW}}_{a}}+\frac{{\mathrm{Wt}}_{b}}{{\mathrm{EEW}}_{b}}}$$

(4)$$\mathrm{NH}\;\mathrm{equivalent}=\frac{\mathrm{Mw}\;\mathrm{of}\;\mathrm{amine}\;\mathrm{curing}\;\mathrm{agent}}{\mathrm{Amount}\;\mathrm{of}\;\mathrm{NH-group}}$$

### 2.4. Characterization of HBE Resins

FT-IR spectra of the synthesized HBE resins were recorded by a PerkinElmer FT-IR System in a wavenumber range of 400–4000 cm^−1^, attenuated total reflectance (ATR) mode and resolution of ±2 cm^−1^. NMR (500 MHz) spectrometer from Varian Unity Inova was used to record the ^1^H NMR and ^13^C NMR spectra of the resins by using CDCl_3_ as solvent and TMS as reference. For degree of branching (DB), the degree of branching of a linear polymer equals 0, while a perfect dendrimer has a DB of 1. DB is the ratio of the sum of integration of dendritic and terminal units to the sum of integration of all repeating units in the structure, measured from ^13^C NMR technique [4,5,8,9], as shown in Equation (5).

(5)$$\mathrm{DB}\;(\%)=\frac{D+T}{D+T+L}\times 100$$

The epoxy equivalent weight (EEW) of the resins was calculated using the standard test methods (ASTM D 1652) [33]. The molecular weight distribution was measured by gel permeation chromatography (GPC), Shimadzu/LC-10ADvp, using a refractive index (RI) detector and CH_3_Cl as mobile phase operated at 40 °C with 1 mL/min. The Mark–Houwink calibration curve correction method was used for standard calibration.

The curing behavior of HBE resins was characterized by differential scanning calorimetry (DSC), DSC 1 STARe Mettler-Toledo, under a nitrogen atmosphere. First, the non-isothermal curing behavior was measured in a range of 25–200 °C and a heating rate of 10 °C/min in order to evaluate the suitable curing temperature [34]. Isothermal curing kinetics was performed at various curing temperatures ranging from 70 to 100 °C. Moreover, the glass transition temperature (T_g_) of the cured epoxy adhesive was also measured by non-isothermal DSC measurement from −30 to 200 °C at a heating rate of 10 °C/min.

### 2.5. Kinetic Analysis

This research determined the kinetic parameters of thermal curing under isothermal condition, which is a conventional method to monitor the curing kinetics [35]. The kinetic parameters included pre-exponential factor (A), activation energy (E_a_), and reaction order (n). 

Both overall heat released and cure rate from heat flow can be measured via DSC. The curing kinetics can be expressed in the following equation:
(6)$$\frac{\mathrm{dQ}}{\mathrm{dt}}{=Q}_{r}\frac{d\alpha }{\mathrm{dt}}{=Q}_{r}k(T)f(\alpha )$$
where dQ/dt is the heat flow, Q_r_ is the total heat released after the reaction was complete, dα/dt is the rate of reaction or curing rate, α is the degree of cure, k(T) is the rate constant, T is the absolute temperature, and f(α) is the reaction model. The degree of cure at time *t* from the isothermal analysis was defined as,
(7)$$\alpha =\frac{H(t)}{H_{T}}$$
when H(t) is the heat of reaction at a certain time *t* and H_T_ is the total heat of reaction. The rate constant can be replaced by an Arrhenius equation. Therefore, Equation (6) can be rearranged as shown in Equation (8):(8)$$\frac{d\alpha }{\mathrm{dt}}=\mathrm{Aexp}(-\frac{E_{a}}{\mathrm{RT}})f(\alpha )$$
where A is the pre-exponential factor, E_a_ is the activation energy, and R is the gas constant (8.314 kJ kmol^−1^ K^−1^).

The kinetic parameters will be meaningless if the reaction model is not suitably used [36]. Generally, three reaction models are classified by characteristic of reaction profile. Vyazovkin et al. recommended how to decide the reaction model by visually inspecting the isothermal reaction profile [37]. The first model is the accelerating model in which the rate increases continuously with rising degree of cure and approaches maximum at the end of the cure state. This type can be explained by a power law model:
(9)$${f(\alpha )=n\alpha }^{(n-1)/n}$$
where n is a constant. The second model is the decelerating model in which the maximum rate is at the initial reaction and it decreases continuously while the degree of cure increases. This type is a common reaction model as expressed in Equation (10):(10)$${f(\alpha )=(1-\alpha )}^{n}$$
where n is the reaction order. The third model is a sigmoidal model in which the rate has the accelerating and decreasing behaviors at the initial and final stages, respectively. This type is the auto-catalyzed reactions, which is known as Šesták–Berggren model [38] as shown in Equation (11):(11)$${f(\alpha )=\alpha }^{m}{\left(1-\alpha \right)}^{n}$$
where n and m are the reaction orders relating to the effects of unreacted reactants and catalytic effect of the product of the reaction, respectively.

Generally, the curing kinetics of the epoxy system can be explained by the autocatalytic model [39] as expressed by Kamal’s equation:(12)$$\frac{d\alpha }{\mathrm{dt}}{=(k}_{1}{(T)+k}_{2}{(T)\alpha }^{m}{)(1-\alpha )}^{n}$$
where k_1_(T) and k_2_(T) are the rate constants and m and n are the reaction orders. When combining Equations (8), (11) and (12) and simplifying the calculations [40], the curing kinetics could best be described by Šesták–Berggren model [41] and the kinetic model is shown in Equation (13).
(13)$$\frac{d\alpha }{\mathrm{dt}}{=k(T)\alpha }^{m}{\left(1-\alpha \right)}^{n}$$
k(T), m and n can be calculated by MATLAB program (version: R2018b) and the activation energy (E_a_) can be determined from taking natural logarithm to Arrhenius’s equation as shown in Equation (14):(14)$$\mathrm{lnk}(T)=\mathrm{lnA}-\frac{E_{a}}{\mathrm{RT}}$$

E_a_ and lnA can be evaluated from the slope and y-intersection of graph plotted between lnk(T) versus 1/T.

## 3. Results and Discussions

### 3.1. Synthesis and Characterization of the Hyperbranched Epoxy Resins

The synthesis of HBE resin began with the polymerization of BPA and PEG using NaOH as base catalyst. The possible synthesized products included diglycidyl ether of polyethylene glycol (DGEPEG), diglycidyl ether of bisphenol A, and diglycidyl ether copolymer of bisphenol A and polyethylene glycol (DGECBAPEG) [42] as displayed in Scheme 1. These in situ products were produced at reaction temperature of 60 °C. When the reaction temperature was heated at 110 °C, A_2_ monomers reacted with pentaerythritol (B_4_) monomer via A_2_ + B_4_ polycondensation reaction to form the hyperbranched epoxy resins, as shown in Scheme 2. Concurrently, epichlorohydrin as epoxidation reagent converted the terminal hydroxyl groups of HBE resins to terminal epoxy groups. The features of the synthesized epoxy resin were investigated by FT-IR and NMR techniques. The FT-IR spectra showed the important functional groups of all resins (Figure 1). There were the stretching vibrations (ν_max_/cm^−1^) of the following feature: 3450 (O–H), 3050 (aromatic C–H), 2970 (aliphatic C-H), 1620 (aromatic C=C), 1249 (C–O), 1040 (C–C), and 915 (oxirane) [4,5]. The FT-IR results of all samples were similar, and it was hardly inspected to identify new chemical bonds. Therefore, the inspection of the chemical bond should be identified via NMR analysis.

The ^1^H-NMR spectra (Figure 2), δ_H_ (ppm), of HBE resin implied the following structural feature: 1.62 (3H, CH_3_), 2.76 and 2.90 (2H, oxirane), 3.38 (1H, oxirane), 3.65 (2H, CH_2_–pentaerythritol unit), 3.70–3.80 (2H, 4CH_2_ of the substituted and unsubstituted pentaerythritol), 3.9 (2H, CH_2_–oxirane), 4.10 (2H, CH_2_–bisphenol-A unit), and 4.15 (1H, OH), 4.20 (1H, CHOH), 6.82 (4H, Ph), and 7.08 (4H, Ph).

The ^1^H-NMR spectra (Figure 3), δ_H_ (ppm), of HBE5P resin implied the following structural feature: 1.62 (3H, CH_3_), 2.78 and 2.90 (2H, oxirane), 3.38 (1H, oxirane), 3.60 (2H, CH_2_-polyethylene glycol), 3.65 (2H, CH_2_–pentaerythritol unit), 3.70–3.80 (2H, 4CH_2_ of the substituted and unsubstituted pentaerythritol), 3.9 (2H, CH_2_–oxirane), 4.08 (2H, CH_2_–bisphenol-A unit), and 4.15 (1H, OH), 4.20 (1H, CHOH), 6.82 (4H, Ph), and 7.08 (4H, Ph).

The ^13^C NMR spectrum (Figure 4), δ_C_ (ppm), of HBE resin implied the following structural feature: 31.0 (CH_3_, bisphenol-A unit), 41.0 (C, isopropylidiene of bisphenol-A unit), 44.0 (CH_2_, oxirane), 44.0–47.0 (central C of pentaerythritol unit), 50.0 (CH, oxirane), 51.0 (CH_2_–oxirane), 62.0–67.0 (CH_2_–O units and CHOH unit), 68.0 (CH_2_, pentaerythritol unit), and 114.0, 127.0, 143.0 and 156.0 (4C, Ph).

The ^13^C NMR spectrum (Figure 5), δ_C_ (ppm), of HBE5P resin implied the following structural feature: 31.0 (CH_3_, bisphenol-A unit), 41.0 (C, isopropylidiene of bisphenol-A unit), 44.0 (CH_2_, oxirane), 44.0–47.0 (central C of pentaerythritol unit), 50.0 (CH, oxirane), 51.0 (CH_2_–oxirane), 62.0–67.0 (CH_2_–O units and CHOH unit), 68.0 (CH_2_, pentaerythritol unit), and 114.0, 127.0, 143.0 and 156.0 (4C, Ph). For the ^1^H-NMR and ^13^C NMR spectra of HBE10P and HBE15P resins, there were the same peak, indicating the important chemical bonding; therefore, this research shows only the spectrum of HBE5P resins.

The degree of branching (DB) of the hyperbranched epoxy resins with various ratios of BPA and PEG was investigated from the ^13^C NMR spectra (Figure 4 and Figure 5) using the four units of central carbon atoms of pentaerythritol [5] (δ_C_(HBE) = 44.9, 45.6, 45.9, and 46.8 ppm and δ_C_(HBE5P) = 44.8, 45.6, 45.8, and 46.7 ppm). The DB was calculated using Equation (5) and the integration values of these peaks were tabulated in Table 2. It was found that all synthesized resins had the hyperbranched structure because DB > 0.5 [9]. The DB values of each formula with and without polyethylene glycol in their structure were hardly different; however, dendritic units decreased and terminal units increased because in situ DGECBAPEG was formed and it might reduce the amount of in situ epoxide group, hindering the generation of branching unit.

Number average molecular weight (M_n_), weight average molecular weight (M_w_) and dispersity (Ð) of the resins are listed in Table 2. The molecular weight of the resin decreased when adding 5 wt % PEG and increased when the PEG amount was further increased (10–15 wt.%), implying an increase of molecular weight. Moreover, the glass transition temperature of the resin decreased with increase of PEG due to the effect of branching density [43] and the internal plasticized effect of PEG.

### 3.2. Curing Behavior of the Hyperbranched Epoxy

The curing study of the hyperbranched epoxy resins cured with diethylenetriamine was investigated by DSC technique. Firstly, curing behavior should be determined to obtain the onset and peak temperatures as well as the heat of reaction of epoxy mixture by non-isothermal DSC method, as shown in Figure 6 and Table 3. Even though chain entanglement occurred in every system, the onset and peak temperatures decreased with increase of PEG in the resins in the case of the resin with 0–10 wt.% PEG because the long-chain structure of PEG acted as plasticizer, increasing the mobility of the polymer chains. However, at 15 wt.% PEG in the resin, the effect of chain entanglement during crosslinking dominated; therefore, the onset and peak temperatures of the resin with 15 wt.% PEG obviously increased [44]. Moreover, the high concentration of PEG chains in the system would delay the curing reaction [45]. Moreover, the heat of reaction increased with increase of PEG due to reduced EEW of the resins in which the active epoxide ring increased. In addition, glass transition temperature (T_g_) of the epoxy thermoset with 0–10 wt.% PEG decreased because of more flexible PEG and an increase in DB [12,13,14], whereas T_g_ of HBE15P thermoset increased exceedingly owing to high crosslink density which could be interpreted from high heat of reaction [46].

### 3.3. Curing Kinetics of the Hyperbranched Epoxy

The curing time of the hyperbranched epoxy at 70, 80, 90, and 100 °C as tabulated in Table 3 was determined by isothermal DSC method. The curing time of all cured hyperbranched epoxy mixtures decreased with increasing curing temperature. This can be attributed to high mobility of epoxy molecules which decreased viscosity, accelerated the rate of cure and reduced the curing time [1].

The kinetic parameters (k, n, and m) were evaluated by fitting the experimental data (cure rate and degree of cure) with Equation (13) via MATLAB program. The results of fitting the data and the equation model are shown in Table 4 and Figure 7. In order to avoid the relative experimental errors for model fitting, the degree of cure should be selected in a range of 0.05–0.95 [37]. It was found that the coefficient of determination (r^2^) of all results was high enough (>0.90), indicating that the experimental data can fit well with the theoretical model in which the rate has the accelerating and decreasing behaviors at the initial and final stages, respectively. It was suggested that the rate constant (k) was a function of curing temperature in which it increased when the temperature increased.

Moreover, the rate constant of the HBE with PEG resins was higher than those without PEG (i.e., HBE) because of a high degree of branching and low entangle structure [47]. The rate constant of HBE5P was the highest, meaning that its cure rate was very fast; however, its curing time was not the lowest because high crosslink network structures slowed down the cure reaction. Furthermore, n order (effect of unreacted materials on the reaction) and m order (catalytic effect of the products on the reaction) of each cured hyperbranched epoxy at the same isothermal temperature were insignificantly different, except HBE5P whose n and m values increased when the temperature increased. The n and m values of HBE5P were higher than those in other systems. It implied that the cure rate of HBE5P is the fastest in the initial stage and then the rate was the slowest at the final stage due to the diffusion control from high crosslink structure [34,48]. In addition, the activation energy of the epoxy at several curing temperatures was calculated from Equation (14) and listed in Table 4. It was found that the activation energy increased with increase of PEG in the resins due to the steric hindrance of PEG structure [49,50], whereas the activation energy of HBE15P decreased because HBE15P had a large amount of equivalent active epoxy group per mass sample (low EEW) which facilitated the curing reaction due to the weakening in the interaction of the molecular chain [17,51].

## 4. Conclusions

In this study, the hyperbranched epoxy resins were synthesized by A_2_ + B_4_ polycondensation reaction with various ratios of polyethylene glycol (PEG) to bisphenol A (BPA). The characterization and feature of synthesized resins were evidently identified by FTIR and NMR analysis. The addition of PEG in the resin enhanced degree of branching, epoxy equivalent weight, and curing reaction. Adding 5–10 wt.% PEG in the resin reduced the onset and peak curing temperatures and glass transition temperature; however, the resin with 15 wt.% PEG showed the increase in these thermal properties due to the lowest epoxy equivalent weight. The curing behavior of all resins followed the auto-catalyzed reaction model (Šesták–Berggren equation). The activation energy increased with increase of PEG in the resins due to the steric hindrance of PEG structure, whereas the activation energy of HBE15P decreased due to a large amount of equivalent active epoxy group per mass sample.

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
