# Peer review of "Synthesis, Thermal Properties and Curing Kinetics of Hyperbranched BPA/PEG Epoxy Resin"

_polymers, 2019, doi:10.3390/polym11101545_

Round 1
Reviewer 1 Report
The article “Synthesis of hyperbranched BPA/PEG Epoxy Resin for electronic application” is a well written and systematic study on the effect of the degree of branching on the physical properties of the resins. Results are well supported by detailed curing reaction kinetics and resins chemical structure studies.
The following is suggested:
The title of the article is misleading. There are no properties analyzed to support that this particular resin is applicable for electronics, such as, mechanical, insulation properties, thermal conductivity, moister protection etc.
I could recommend to modify the title to highlight the aspects that were studied e.g. curing kinetics.
Line 121: please specify how the active functional groups amount was determined.
For GPC: please specify standards used for calibration and chromatography columns separation range (cutoffs), flow used and temperature.
Line 239 and Table 2: Please, replace PDI and polydispersity index in the text according to the IUPAC recommendations (dispersity, represented by the symbol Đ ) see Pure Appl. Chem., Vol. 81, No. 2, pp. 351–353, 2009.
Line 95: potassium acid phthalate. Please use IUPAC naming it should be Potassium hydrogen phthalate
Line 121: HPE. Shouldn’t it be HBE?
Line 219-222 and Line 231: 13C NMR signals should be given with one decimal figure.
Table 2: units for Mw and Mn are missing
Reviewer 2 Report
This work study the synthesized of Hyperbranched BPA/PEG Epoxy Resin via 10 A2 + B4 polycondensation reaction at various BPA/PEG ratios. The approach such as NMR and results are sufficiently described, but the explanation from the results in the paper is from a scientific point of view insufficiently. Here my some comments:
The authors given the FT-IR and NMR results of the synthesized epoxy resin. However, the same and different are not given. For example, in Figure 1, the authors only given the functional groups. Therefore, more explanations should be given. “the onset and peak temperatures of the resin with 15wt% PEG obviously increased due to the effect of chain entanglement during crosslinking”. Why there is not chain entanglement for epoxy thermoset with 0 -10 wt% PEG. The title mentioned the electronics application, however the reviewer can not found. Please give the detail in the Text. Suggest to cite two references. [Polymer Testing, 2019, 75, 367; Progress in Organic Coatings 2018, 122, 219]
